# A Machine Learning Model for Predicting Unscheduled 72 h Return Visits to the Emergency Department by Patients with Abdominal Pain

**DOI:** 10.3390/diagnostics12010082

**Published:** 2021-12-30

**Authors:** Chun-Chuan Hsu, Cheng-C.J. Chu, Ching-Heng Lin, Chien-Hsiung Huang, Chip-Jin Ng, Guan-Yu Lin, Meng-Jiun Chiou, Hsiang-Yun Lo, Shou-Yen Chen

**Affiliations:** 1Department of Emergency Medicine, Chang Gung Memorial Hospital and Chang Gung University, Linkou, Taoyuan City 333, Taiwan; b97401074@cgmh.org.tw (C.-C.H.); ngowl@ms3.hinet.net (C.-J.N.); 2Center for Artificial Intelligence in Medicine, Chang Gung Memorial Hospital, Linkou, Taoyuan City 333, Taiwan; glacrimosa@gmail.com (C.-C.J.C.); chingheng113@gmail.com (C.-H.L.); 407336038@gapp.fju.edu.tw (G.-Y.L.); mengjun89@gmail.com (M.-J.C.); 3Bachelor Program in Artificial Intelligence, Chang Gung University, Taoyuan City 333, Taiwan; 4New Taipei City Hospital, New Taipei City Government, New Taipei City 241, Taiwan; innoto86@gmail.com; 5Graduate Institute of Clinical Medical Sciences, Division of Medical Education, College of Medicine, Chang Gung University, Taoyuan City 333, Taiwan

**Keywords:** unscheduled return visit, 72 h, emergency department, abdominal pain

## Abstract

Seventy-two-hour unscheduled return visits (URVs) by emergency department patients are a key clinical index for evaluating the quality of care in emergency departments (EDs). This study aimed to develop a machine learning model to predict 72 h URVs for ED patients with abdominal pain. Electronic health records data were collected from the Chang Gung Research Database (CGRD) for 25,151 ED visits by patients with abdominal pain and a total of 617 features were used for analysis. We used supervised machine learning models, namely logistic regression (LR), support vector machine (SVM), random forest (RF), extreme gradient boosting (XGB), and voting classifier (VC), to predict URVs. The VC model achieved more favorable overall performance than other models (AUROC: 0.74; 95% confidence interval (CI), 0.69–0.76; sensitivity, 0.39; specificity, 0.89; F1 score, 0.25). The reduced VC model achieved comparable performance (AUROC: 0.72; 95% CI, 0.69–0.74) to the full models using all clinical features. The VC model exhibited the most favorable performance in predicting 72 h URVs for patients with abdominal pain, both for all-features and reduced-features models. Application of the VC model in the clinical setting after validation may help physicians to make accurate decisions and decrease URVs.

## 1. Introduction

The 72 h unscheduled return visit (URV) rate has been used for decades to evaluate the quality of care in hospital emergency departments (EDs) [1,2,3]. Among ED patients, presenting with abdominal pain is a risk factor for URVs [4,5,6,7]. Abdominal pain is a common chief complaint among ED patients, and making a diagnosis for patients with abdominal pain is challenging for emergency physicians because the pain’s etiology can be diverse [8,9,10]. Causes of abdominal pain range from mild abdominal infection, such as acute gastroenteritis, to life-threatening emergencies, such as ischemic bowel, and are occasionally even undifferentiable. In a 35-year retrospective study, up to 21.1% of patients had undifferentiable abdominal pain when the disposition was made [11]. Uncertain or missed diagnoses can result in inappropriate dispositions, thus increasing medical expenses and patient mortality [12].

Machine learning is a popular research field in medicine, and the application of the technology is now frequently discussed [13,14]. Machine learning can incorporate a variety of variables to make classifications or predictions related to complicated problems [15]. Because of these characteristics, machine learning can assist physicians in multiple clinical situations, such as image study classification and mortality prediction [16,17]. Previous studies have attempted to use machine learning models to predict the 72 h URVs of ED patients [18,19]. These previous models attempted to predict URVs for all causes, which is a highly complex task and may have reduced the accuracy of prediction. Focusing on a specific but crucial group of ED patients such as those with abdominal pain may increase the accuracy of predictions together with their practical applications. In addition to developing the prediction model, machine learning can help identify previously unknown essential features of URVs [20,21], and a reduction in URVs may be achieved by improving the factors found through such an analysis.

In the present study, we aimed to use machine learning to develop a practical prediction model for 72 h URVs by ED patients with abdominal pain. Multiple models of machine learning were examined and compared to determine the most feasible and precise model. Our secondary goal was to identify possible meaningful clinical indicators affecting URVs to improve quality of care.

## 2. Materials and Methods

### 2.1. Study Design

This was a retrospective study using patients’ electronic health record (EHR) data from Chang Gung Memorial Hospital (CGMH) to predict 72 h ED URVs. The study focused on a specific cohort of ED patients with the chief complaint of abdominal pain, which is one of the primary causes of URVs in the ED. We included all adult patients (age >18 years) coming to the ED at Taipei and Linkou CGMH from 1 January 2018 to 31 December 2019. Exclusion criteria were patients who had scheduled return visits, who died after index ED visits, who were transferred to another hospital, or who were discharged against medical advice.

A total of 617 clinical features were collected from the data, including demographics, mode of arrival to the ED, time of arrival, seniority of the primary ED physician, and clinical factors such as length of stay (LOS), triage vital signs, blood tests, exams, and drugs used at index ED visits. Diagnosis-related features were formulated as the first three numbers of the International Classification of Diseases, Tenth Revision (ICD-10) code from patient medical histories derived from EHRs. A variable called “free typing” was also included in this study. It is a binary variable indicating whether or not there is a free form order. A free typing order in our system was a blank space that physicians could type any additional order or notice in our system. Other features included frequency of visiting the ED within the previous year, frequency of hospital admission within the previous year, and primary diagnoses of diseases within the previous two years.

### 2.2. Data Source and Preprocessing

The EHR data were collected from the Chang Gung Research Database, which is the largest multi-institutional EHR database in Taiwan, comprising seven institutions of CGMH [22]. The study protocol was approved by the Institutional Review Board of CGMH (IRB No. 202000624B0, passed on 8 April 2020). All patient information included in this study was anonymized and deidentified.

For physiological measurements (e.g., vital signs), we first replaced extreme values from the empirical distribution with missing values. These values may represent human error instead of true measurements and should be treated as missing data (e.g., weight >400 kg). For continuous variables, we defined the outliers to be values greater or smaller than 1.5 times the interquartile range of the empirical distribution. The outlier data were also replaced with missing values for later data imputation. For discrete variables, the original missing values were treated as a new category, and if the frequency of a particular category was less than five visits, we removed all the visits associated with that category. This was to ensure that enough samples were available to be split for model training, model validation, and model testing. There are only four variables have significant missingness (>10%), and they are the first and last measurement of SBP and DBP in the ER. The missing percentage is around 33%. Triage blood pressure measurement was recorded as another variable and the number of missing is insignificant (<10%). We examined those four variables and found that the patients with relatively mild disease (triage level 3, 4; consciousness clear; no pain killer used) tend to have those four missing variables, and missing not at random (MNAR) could possibly exist. To further control this factor, we tried senMice imputation and found comparable results to the median imputation, so the results from median imputation were adopted in this study. We then split the data for training and testing in an 80 to 20 ratio. The test set was stored after the train–test split and tested after the final model was created. The missing data for the continuous variables were imputed with the median value of the distribution. If the values for a particular feature were missing for more than half of the total visits, the feature was discarded. The imputation of 20% holdout data (the test data) was based on the median value derived from the testing data. All features were then encoded before being fit to the model, and the data for continuous variables were standardized.

### 2.3. Machine Learning Model and Training

For data imbalance between cases with and without URVs within 72 h (the ratio was approximately 19:1), a one-sided selection algorithm was applied. The one-sided selection algorithm yielded a corrected ratio of data for cases with and without URVs within 72 h of discharge from the ED (approximately 15:1), and samples around the decision boundaries were removed by applying a Tomek link algorithm and then a condensed nearest neighbor algorithm (with k = 1). We also experimented with a range of random seed values from 50 to 700 in 50-step increments to determine whether balancing the ratio would improve final performance. However, no further improvements were observed, and we thus fixed the random seed value at 50. The model we used included logistic regression (LR), random forest (RF), extreme gradient boosting (XGB), and voting classifiers (VC); VC was used to make the final prediction from the previous three models. We trained each model in 10-fold cross-validation to select the best model for the final testing of the 20% holdout data. To further control data imbalance during model training, we applied the bootstrapping method (sample with replacement) to balance the frequency of cases with and without URVs. Bootstrapping was applied each time when training the model but not when validating the model. Calibration was done and Brier score was also calculated.

### 2.4. Hyperparameter Tuning and VC

We systematically performed hyperparameter tuning for each classifier by using different mechanisms. For LR, we used a grid search algorithm to test combinations of hyperparameters, which includes solvers and c-values. For RF, we used a random search algorithm because the complexity of the decision tree was higher than that for LR. The hyperparameters in RF are max_depth, max_features, min_samples_leaf, min_samples_split, and n_estimators. For XGB, we tuned parameters on the basis of Bayesian optimization by using hyperopt packages. Detailed descriptions of hyperparameters are described in Appendix A. We built the ensemble classifier to improve the generalizability of single classifiers by adopting soft-voting strategies. Soft voting took the predicted probability from each classifier and then weighted each classifier by importance to yield the final prediction about the label. We completely tested 33 combinations of voting weights among three classifiers (assigning the importance of each classifier range from 1 to 18) and then selected the voting weight that yielded the most favorable performance (Appendix A).

### 2.5. Reduced-Features Models

We selected the top 10 features for each of the three classifiers ranked by their importance and identified those in common among these 30 top features. LR feature importance was based on the absolute value of the beta coefficient of the regression. For RF, feature importance was based on a mean decrease in Gini impurity across all splits, and the importance of XGB features was based on weighting, which was defined as the number of times a feature was used to split the data across all trees. We repeated the same process for model training, parameter tuning, and voting, and conducted the same analysis for the reduced-features models as for the all-features models.

### 2.6. Outcome Measurement and Statistical Analysis

The features and variables were analyzed using SPSS software (version 13.0 for Windows; SPSS, Chicago, IL, USA). The Wilcoxon rank-sum test was applied to examine the significant differences among features that were continuous variables. For those features that were discrete or categorical, the chi-square test was applied.

For each classifier, we calculated several metrics to evaluate and compare performance between models, including the area under the receiver operating characteristic (ROC) curve (AUC), accuracy, sensitivity, specificity, precision, F1 score, and confusion matrix. All metrics were implemented with the Python scikit-learn package (version 1.0.1). Accuracy indicated the number of cases of URV and non-URV that were correctly identified. Sensitivity indicated the number of cases that were correctly identified as true among the cases identified as URV. Specificity indicated the number of cases that were correctly identified as false among the cases identified as non-URV. Precision indicated the number of cases that were correctly identified among the true cases. The F1 score was the harmonic mean of sensitivity and precision values, which represented the balance between the two indices. All models were compared with the AUC and the noted metrics by evaluating the holdout data.

## 3. Results

A total of 449,594 eligible ED visits by 290,914 patients occurred during the study period from 1 January 2018 to 31 December 2019 (Figure 1). A total of 266,620 visits were excluded for patients aged <18 years old. Patients with trauma, patients discharged against medical advice, patients with a scheduled return visit, or a return visit occurring after 72 h were also excluded. Among the 183,334 ED visits after exclusion, 25,151 ED visits were associated with abdominal pain and included in our study. Of these visits, 1471 discharges (5.85%) had a 72 h return visit, and the ratio accorded with previous studies [23]. Eighty percent (20,120) of the included visits were distributed as training sets for machine learning, and the remaining 20% were used as testing sets.

### 3.1. Characteristic Description

The demographics and major characteristics of the cohort are listed in Table 1. The patients who had 72 h URVs were older (52.13 vs. 46.44 years, *p* < 0.001), and more often male (49.8% vs. 41.0%, *p* < 0.001) compared to those without URVs. These patients also had more ED visits in the previous year (1 vs. 0, *p* < 0.001). No significant difference in the method of arrival or the LOS between groups was observed, but a lower proportion of triage levels 1 and 2 was noted in the patients with return visits (5.3% vs. 4.1%, *p* = 0.013). For patient vital signs, the patients with 72 h URVs had higher systolic blood pressure (136 vs. 131 mmHg, *p* < 0.001) and diastolic blood pressure (83 vs. 80 mmHg, *p* = 0.005). No significant differences in examinations were observed between these two patient groups except for blood tests, which were more common in patients with URVs (57.2% vs. 54.0%, *p* = 0.002).

### 3.2. Performance of the All-Features Models

Multiple prediction models, including LR, RF, XGB, and VC models, were tested and compared in our study. The most favorable voting weight of the VC for the prediction model was 1(LR):2(RF):1(XGB) (Figure 2). Calibration result was showed as calibration curve (Appendix A). Brier score was calculated and the results were revealed as LR: 0.19, RF: 0.18, XGB: 0.05, and VC: 0.16. Calibration curve showed XGB have better probability prediction and Hosmer–Lemeshow test for goodness of fit confirmed that XGB has much better fit comparing to other classifiers (LR: *p* < 0.0001, RF: *p* < 0.0001, XGB: *p* = 0.03, VC: *p* < 0.0001). The predictive performances for each model are listed in Table 2. Models were trained in all features, and all models were able to perform 72 h URV prediction with an AUC greater than 0.69. The XGB and VC models achieved the most favorable results, with test AUCs of 0.74 (95% CI, 0.7–0.76) and 0.74 (95% CI, 0.69–0.76), respectively. Additionally, VC exhibited higher F1 scores (0.25) than XGB (0.07). LR had a test AUC of 0.73 (95% CI, 00.7–0.76), and RF had a test AUC of 0.71 (95% CI, 0.69–0.75). XGB exhibited the highest specificity (0.99) and precision (0.92) but the lowest sensitivity (0.04). LR exhibited the highest sensitivity (0.59) but the lowest specificity (0.76) and precision (0.13). Among these models, VC had the most favorably balanced comprehensive performance.

### 3.3. Reduced-Features Models Performance

The reduced-features models used the top 10 features from each model (Figure 3). The performance results of the reduced-features models are listed in Table 3. Reduced XGB exhibited a mildly higher AUC (0.73; 95% CI, 0.68–0.75) than reduced VC (0.72; 95% CI, 0.69–0.74) but a lower F1 score (0.07 vs. 0.24). Similar trends in the performance of all-features models were noted in the reduced-features models. Reduced XGB exhibited the highest specificity (0.99) and precision (0.91), and reduced LR exhibited the highest sensitivity (0.54). Reduced VC still exhibited relatively balanced performance in terms of sensitivity (0.39), specificity (0.88), and precision (0.17).

### 3.4. Comparison of All-Features and Reduced-Features Models

The AUCs for both all-features and reduced-features models are displayed in Figure 4. Reduced-features models exhibited slightly lower AUCs than did all-features models. VC and XGB using all features had the highest AUCs (0.74). Comparisons of precision–recall curves for different models were performed and indicated tradeoffs in limitations and precision–recall for both all-features and reduced-features models (Figure 5).

## 4. Discussion

To our knowledge, this study is the first to examine the use of machine learning in the prediction of 72 h URVs for patients with abdominal pain presenting to the ED. Previous studies have reported prediction models using machine learning to predict 72 h URV for all causes, but no previous studies have focused on ED patients with abdominal pain [18,19,24]. One study examined the relationship between triage level and return visits in patients with abdominal pain, but the study lacked a prediction model for these patients [25]. Although predicting 72 h URVs for ED patients is considered crucial to improving the quality of care and patient safety, a 72 h URV is generally considered a complex and multicause event [2,3]. One difficulty in predicting 72 h URVs for ED patients is the complexity of the chief complaints and diseases. We attempted to focus on one specific group of ED patients to develop a better prediction result, but the results approximated those of previous studies. A possible explanation is the different medical approaches and culture in our country. Patients may not return to the same hospital after their first visit because the accessibility of different hospitals in our country is high [26]. However, we were able to develop a practical model for predicting URVs for ED patients with abdominal pain by using complex, nonlinear machine learning models.

LR is the most commonly used classifier in clinical practice [27,28,29]. However, relationships between 72 h URVs and the variables may not be linear; therefore, adopting a nonlinear model was necessary. RF was used in our study because it is a robust classifier that is widely used [30,31]. XGB was also introduced in our study because of favorable performance with imbalanced data due to autoregulating class weight during training [32,33]. In our training process, we found that some classifiers exhibited false positive and false negative preferences (e.g., LR tended to exhibit false positive prediction, whereas for XGB, the tendency was for false negative prediction). For better prediction ability, a VC combining these three models was used, and the most favorable weighting for these three models was obtained after multiple iterations. The observation that VC outperformed other individual classifiers is not surprising [34]. Tradeoffs between precision and recall were observed in all prediction models in our study. Using a classifier with high recall could catch most high-risk unscheduled-return patients, but it may also cause patient waiting and ED crowding. Classifiers with high precision can avoid this situation by lowering the number of false alarms; yet, they might miss more 72 h URVs. Our final prediction model using VC with a specific weighting ratio achieved a relatively favorable balance. However, further validation of the prediction model is necessary to achieve the most favorable balance.

We extracted the top 10 factors of each model through feature importance analysis and made a reduced-features model that had a noninferior result compared with the model using all features. Developing a reduced-features model offered many advantages, including easier application and lower demand for computing power. Extracted features also provided clinical insights. In addition to previously discovered risk factors, such as the number of visits to the emergency department in the previous year and previous month, age, and LOS, some additional essential factors were found in our analysis [5,35]. Patients with a record of “free typing” were found to have higher URV rates. In the clinical setting of our hospital, we use “free typing” to record conditions such as refusing hospitalization, expressing hostility to healthcare providers, and other psychological factors and social situations. In this study, machine learning identified this variable as a 72 h URV risk factor. Patients with this risk factor tended to exhibit more complex medical, psychological, and social factors that affected clinical decision-making; the risk of 72 h URV was therefore higher [36]. The use of the analgesic ketorolac, higher systolic blood pressure at triage, and larger numbers of X-ray inspections during the previous two years were also important features associated with prediction. Previous research has revealed that the use of analgesia would not mask the warning signs or delay the diagnosis of abdominal pain in patients [37,38,39]. However, the use of ketorolac and higher systolic blood pressure may indicate that a patient is experiencing more severe pain, which is a proven risk factor for URVs [40]. A greater number of X-rays taken in a given time period may imply that the patient’s condition is more complicated and that the patient probably has other underlying conditions. For these patients with comorbidities, dispositions should be made carefully after thorough deliberation [19,41]. According to our study, close observation of the symptoms or the arranging of further exams should be considered for these patients. Further research is still warranted to clarify the role of the additional factors found in our study.

Using machine learning to assist physicians offers several benefits, such as improving clinician performance in diagnosis, clinical workflow optimization, and even the assessment of physician competence [42,43,44]. Applying our prediction model may improve patient safety and decrease URVs to the ED. This prediction model, as an alarm system, may remind the physician to consider thorough survey before making the discharge decision for the patient with high risk of URV. However, unnecessary laboratory and imaging studies may be ordered due to false positive predictions [45]. Accordingly, LOS for these patients and ED crowding may both increase, although LOS could be affected by other hospital related factors. Machine learning models should be applied carefully in the real world, and the accuracy of such models may not be as favorable as in a retrospective study [46,47]. Further validation studies are needed to evaluate the effects of implementing such a prediction model in clinical settings, including the effects on patient safety, 72 h URV rate, and ED crowding.

### Limitations

Our research has several limitations. This study predicted 72 h URVs but not the disposition or severity of the condition after the return visit because of a shortage of and imbalance in the data. The complexity and quantity of the data can still be improved, and further analysis of the prognoses should be conducted after including more data. Additionally, the imaging results and the records of the ED nurses were not included in the study because of technical problems, and this may decrease the accuracy of our prediction model. Finally, our research examined a single medical institution and lacked data from other hospitals. Patients with visits to other hospitals within 72 h could not be included in our analysis, which may have affected the models’ predictive ability. Multicenter involvement could improve the data quality and achieve more favorable performance for the prediction model.

## 5. Conclusions

We developed a practical machine learning model by using VCs with specific weighting for LR, RF, and XGB. The VC prediction model had the most favorable balance and performance compared with other machine learning models. A reduced-features model was also developed and had noninferior performance, which may be more useful in clinical workflow because of lower demands for data and computing power. Some key features associated with URVs were also identified and can be applied in the clinical setting to improve quality of care and avoid return visits of ED patients with abdominal pain. Further study is necessary to validate the prediction model and evaluate the impact of its application.

## Figures and Tables

**Figure 1 diagnostics-12-00082-f001:**
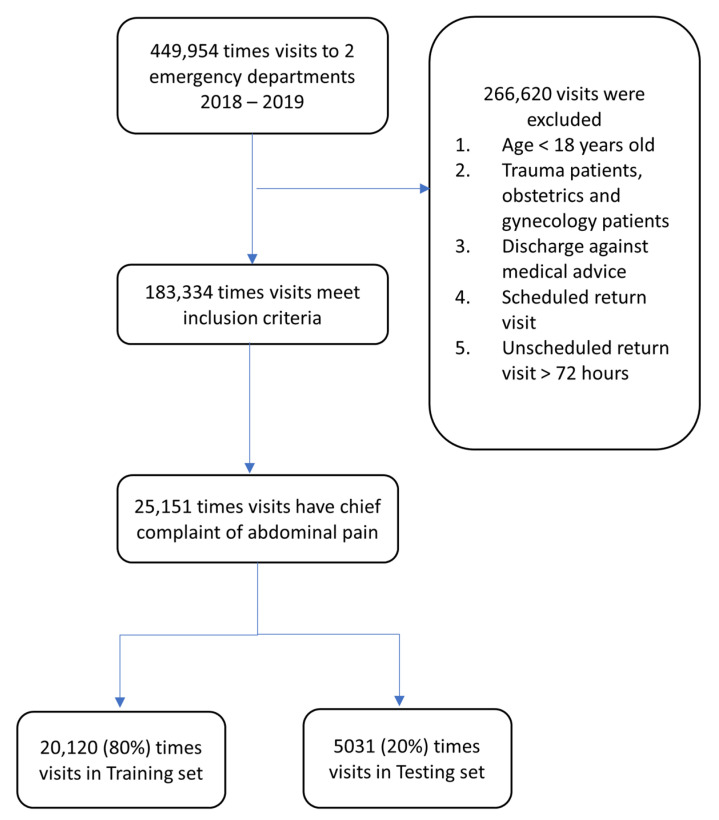
Study population and split of training and testing sets.

**Figure 2 diagnostics-12-00082-f002:**
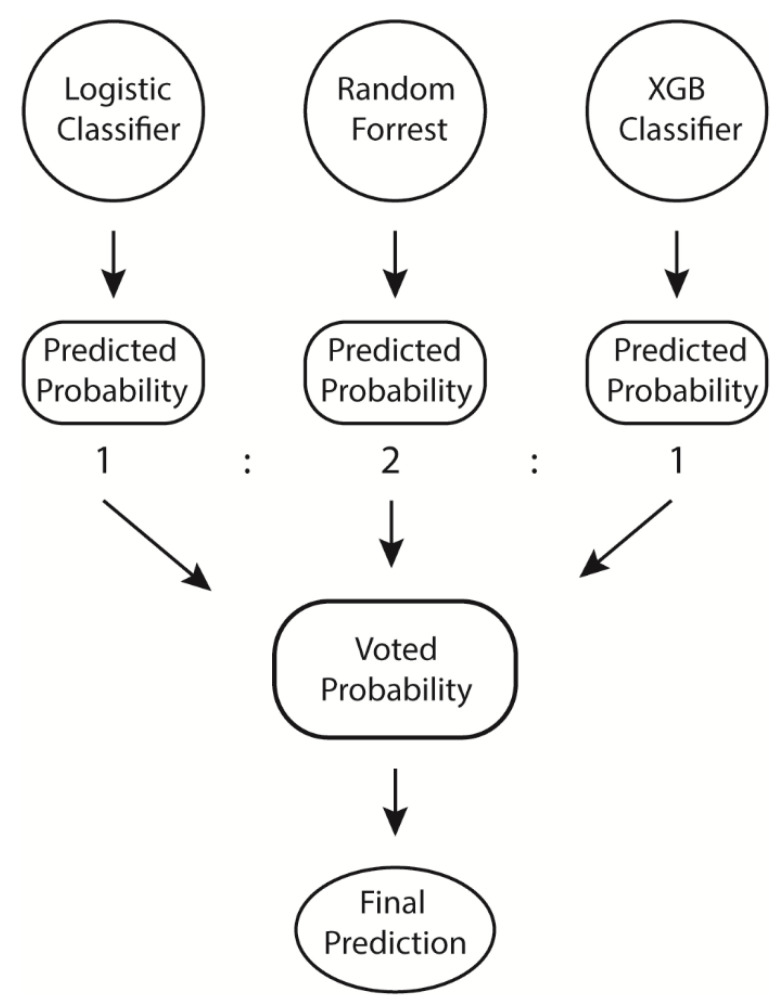
Voting classifier structure and weight of each model.

**Figure 3 diagnostics-12-00082-f003:**
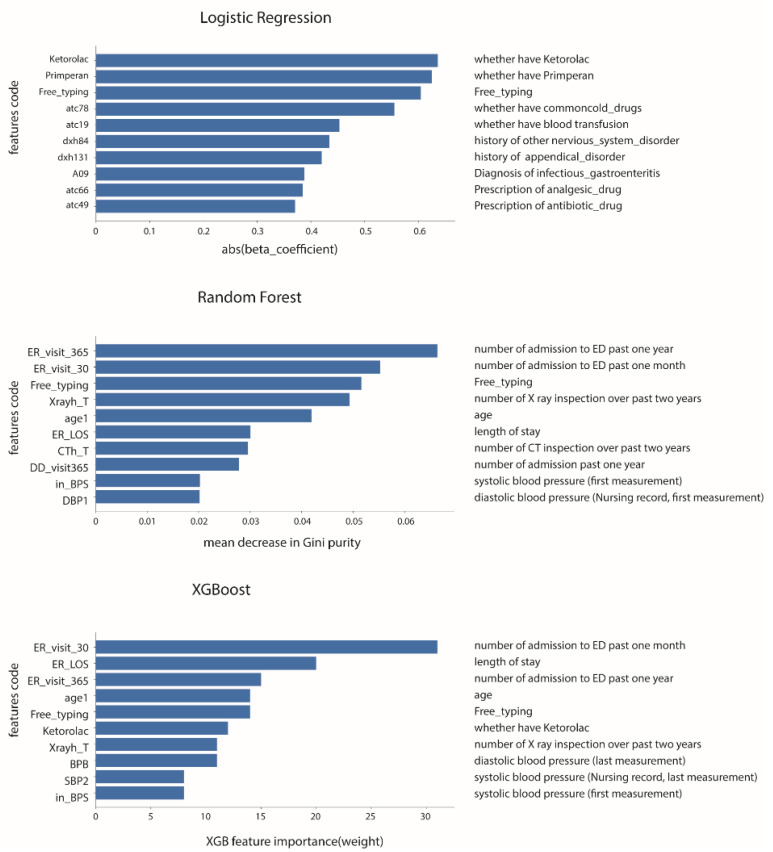
Top 10 important features extracted by each model.

**Figure 4 diagnostics-12-00082-f004:**
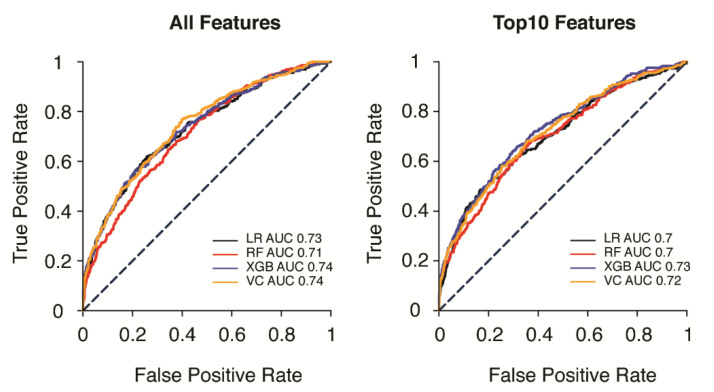
Comparisons of performance (AUC) for different models in predicting unscheduled emergency department revisits within 72 h of discharge: (**left**) prediction made by using all collected features; (**right**) the same prediction made using only the top 10 features from each model (ranked by importance).

**Figure 5 diagnostics-12-00082-f005:**
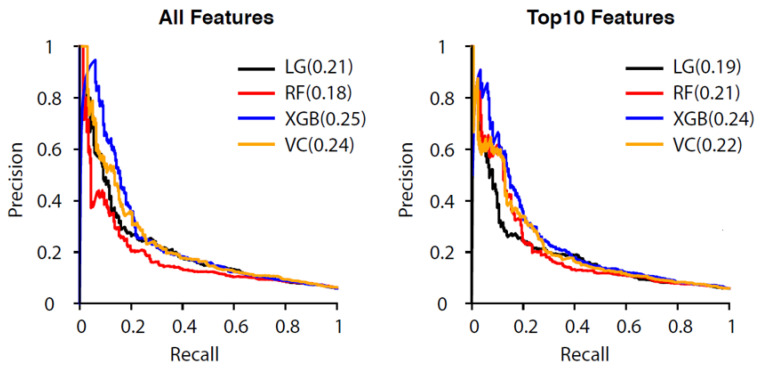
Comparisons of the precision–recall curves (PR curves) of different models in predicting unscheduled revisits to the emergency department within 72 h of discharge: (**left**) prediction made by using all collected features; (**right**) the same prediction made using only the top 10 features from each model (ranked by importance). The area under the PR curve is shown in parentheses.

**Table 1 diagnostics-12-00082-t001:** Summary statistics for demographic and clinical features of the data.

	Training Set		Testing Set		All Encounters		
	No 72 h Return Visit N = 18,943	72 h Return Visit N = 1177	No 72 h Return Visit N = 4737	72 h Return Visit N = 294	No 72 h Return Visit N = 23,680	72 h Return Visit N = 1471	*p*-Value
**Demographic** Age, Mean (SD), years	46.44 (18.12)	52.15 (18.22)	46.44 (18.26)	52.02 (18.31)	46.44 (18.15)	52.13 (18.23)	<0.001
Male, No. %	7811 (41.23%)	567 (48.17%)	1897(40.05%)	165 (56.12%)	9708 (41.0%)	732 (49.8%)	<0.001
**ED related features** Arrival by ambulance, No. %	140 (0.74%)	7 (0.59%)	36 (0.76%)	3 (1.02%)	176 (0.7%)	10 (0.7%)	0.255
Previous ED visits in the past year, Median (IQR)	0 (0–1)	1 (0–3)	0 (0–1)	1 (0–3)	0 (0–1)	1 (0–3)	<0.001
Triage level > 3, No. %	959 (5.07%)	50 (4.25%)	295 (6.23%)	10 (3.4%)	1254 (5.3%)	60 (4.1%)	0.013
Length of stay, minutes, Median (IQR)	106.2 (67.2–198)	115.2 (75–193.8)	103.8 (64.2–190.8)	115.8 (73.4–197.7)	106.2 (66–196.8)	115.2 (74.4–196.5)	0.237
**Vital signs** Body temperature at triage, Median (IQR)	36.3 (35.9–36.7)	36.3 (35.9–36.8)	36.3 (36–36.8)	36.3 (35.8–36.7)	36.3 (35.9–36.8)	36.3 (35.9–36.8)	0.066
Heart rate at triage, Median (IQR)	83 (73–94)	83.5 (73–96)	83 (73–95)	83 (71–95)	83 (73–95)	83 (73–96)	0.113
Respiratory rate at triage, Median (IQR)	18 (17–19)	18 (17–19)	18 (17–18)	18 (17–19)	18 (17–18)	18 (17–19)	<0.001
Systolic blood pressure, Median (IQR)	131 (116–149)	136 (120–155)	131 (116–149)	135.5 (119–153)	131 (116–149)	136 (120–155)	<0.001
Diastolic blood pressure, Median (IQR)	80 (70–90)	83 (72.2–93)	80 (69–90)	82 (71–90)	80 (70–90)	83 (72–93)	0.005
**Examinations** Blood test, No. %	10,251 (54.11%)	677 (57.52%)	2539 (53.6%)	164 (55.78%)	12,790 (54.0%)	841 (57.2%)	0.020
X-ray, No. %	9794 (51.7%)	635 (53.95%)	2411 (50.9%)	147 (50%)	12,205 (51.5%)	782 (53.2%)	0.238
Abdominal echo, No. %	391 (2.06%)	18 (1.53%)	105 (2.22%)	6 (2.04%)	496 (2.1%)	24 (1.6%)	0.264
CT, No. %	2565 (13.54%)	143 (12.15%)	626 (13.22%)	41 (13.95%)	3191 (13.5%)	184 (12.5%)	0.309

Abbreviations: SD, standard deviation; ED, emergency department; IQR, interquartile range; CT, computed tomography.

**Table 2 diagnostics-12-00082-t002:** All-features model performance for predicting unscheduled return visits within 72 h by patients with abdominal pain.

Model Name	Accuracy	AUC	Sensitivity	Specificity	Precision	F1 Score
LR	0.75	0.73 (0.7–0.76)	0.59	0.76	0.13	0.22
RF	0.85	0.71 (0.69–0.75)	0.33	0.88	0.14	0.20
XGB	0.94	0.74 (0.7–0.76)	0.04	0.99	0.92	0.07
VC	0.86	0.74 (0.69–0.76)	0.39	0.89	0.18	0.25

Abbreviations: LR, logistic regression; RF, random forest; XGB, extreme gradient boost; VC, voting classifier combination; AUC, area under the receiver operating characteristic curve. LR, RF, and XGB: parenthetical indicates (5%, 95%) confidence interval estimated through the bootstrapping method.

**Table 3 diagnostics-12-00082-t003:** Reduced-features model performance for predicting unscheduled return visits within 72 h by patients with abdominal pain.

Model Name	Accuracy	AUC	Sensitivity	Specificity	Precision	F1 Score
LR	0.74	0.70 (0.68–0.73)	0.54	0.75	0.12	0.19
RF	0.87	0.70 (0.68–0.73)	0.31	0.91	0.17	0.22
XGB	0.94	0.73 (0.68–0.75)	0.03	0.99	0.91	0.07
VC	0.85	0.72 (0.69–0.74)	0.39	0.88	0.17	0.24

Abbreviations: LR, logistic regression; RF, random forest; XGB, extreme gradient boost; VC, voting classifier combination; AUC, area under the receiver operating characteristic curve. LR, RF, and XGB: parenthetical indicates (5%, 95%) confidence interval estimated through the bootstrapping method.

## Data Availability

The data that support the findings of this study are available from Linkou Chang Gung Memorial Hospital, but restrictions may apply to the availability of these data, which were approved by individual hospital IRB for the current study, and thus not publicly available. However, processed datasets can be requested and made available from the authors with the permission of Linkou Chang Gung Memorial Hospital.

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
