# Peer review of "A Machine Learning Model for Predicting Unscheduled 72 h Return Visits to the Emergency Department by Patients with Abdominal Pain"

_diagnostics, 2021, doi:10.3390/diagnostics12010082_

Round 1

Reviewer 1 Report

Overall, I applaud the authors in preparing this manuscript and investigating the important and interesting topic on unscheduled return visits (URV) in patients with abdominal pain. Abdominal pain is amongst the most common reason for patients to present to an Emergency Department and have a wide range of underlying diseases, thus challenging regarding disposition decision making. Prediction of unscheduled return visits can help disposition choices. The generally well written, the introduction should include the implication of predicting URV more clearly. Please find my comments below. 

Major:

The authors used simple median imputation method for all missing variables, which has several limitations per se: e.g. underestimated variances. These could be addressed with multiple imputation methods. Additionally, and more importantly, single median imputation assumes no MNAR, however, EHR data tends to have a lot if MNAR data (see for example Haneuse S, Arterburn D, Daniels MJ. Assessing Missing Data Assumptions in EHR-Based Studies: A Complex and Underappreciated Task. JAMA Netw Open. 2021;4(2):e210184. doi:10.1001/jamanetworkopen.2021.0184), which should be addressed.

The authors do not mention a correlation analysis for the used features, which can lead to skewed results in their logistic regression (LR) model. This could be the case if the likely correlated data e.g. “systolic blood pressure (Nursing record, last measurement)” and “systolic blood pressure (first measurement)” used in XGB model is used in the LR model

The described model mainly uses only patient-related data and no hospital system and just one provider related factors. Using this approach can be beneficial since data collection is based on individual patients’ data only. However, patients LOS can be confounded by hospital-related factors like overcrowding, thus therefore is not a simple patient-related data point.

While I agree with the authors that prediction of URV can improve patients’ safety and decrease URVs, it’s not clear to me how prediction of URV can “avoid delays in diagnosis”.

The authors state in the discussion that “It may also help physicians identify higher-risk patients with abdominal pain and arrange examinations earlier to confirm diagnoses or indicate appropriate outpatient department follow-up.” However, prediction of URV do not necessarily help to distinguish of URV occurred due to misdiagnosis or lack of ED discharge planning. To address this issue, the models would have to include a more differentiated assessment of URV including severity and causation of URV. While this is an interesting topic this seems to be out of scope for this article, however, the discussion should clearly distinguish that the described model can be the base for this aim, however, cannot provide this at the current state.

Minor:

The term “free typing” is first explained in the discussion, but used within the results (e.g. Figure 3). A definition should be placed in the methods sections

Figures 1, 2 are blurred

Table 3 The term “AUROC” is used in the title, however in the Abbreviation only the term “AUC” is used

Author Response

Reviewer 1

Overall, I applaud the authors in preparing this manuscript and investigating the important and interesting topic on unscheduled return visits (URV) in patients with abdominal pain. Abdominal pain is amongst the most common reason for patients to present to an Emergency Department and have a wide range of underlying diseases, thus challenging regarding disposition decision making. Prediction of unscheduled return visits can help disposition choices. The generally well written, the introduction should include the implication of predicting URV more clearly. Please find my comments below. 

Major:

The authors used simple median imputation method for all missing variables, which has several limitations per se: e.g. underestimated variances. These could be addressed with multiple imputation methods. Additionally, and more importantly, single median imputation assumes no MNAR, however, EHR data tends to have a lot if MNAR data (see for example Haneuse S, Arterburn D, Daniels MJ. Assessing Missing Data Assumptions in EHR-Based Studies: A Complex and Underappreciated Task. JAMA Netw Open. 2021;4(2):e210184. doi:10.1001/jamanetworkopen.2021.0184), which should be addressed.
ANS: Thank you for your suggestion. We have tried different imputation methods. Different methods have yield comparable results of the original method. Imputation by mean performance for each classifier (AUC/f1 score): logistic regression(0.73/0.21)random forest(0.71/0.22)extreme gradient boosting(0.72/0.07)voting classifier(0.73/0.25)). Imputation by most frequent value: logistic regression(0.73/0.21)random forest(0.72/0.21)extreme gradient boosting(0.73/0.06)voting classifier(0.73/0.25)).

The authors do not mention a correlation analysis for the used features, which can lead to skewed results in their logistic regression (LR) model. This could be the case if the likely correlated data e.g. “systolic blood pressure (Nursing record, last measurement)” and “systolic blood pressure (first measurement)” used in XGB model is used in the LR model
ANS: Thank you for your comment. We have done the correlation analysis of all continuous variables and the result was provided as supplementary file 1. We noticed that most of the features which are highly correlated belong to the category of vital sign, especially between first measurement and last measurement. The other category is the triage index. In order to know whether the correlation have profound impact on our model performance, we re-run the model, which removed the last measurement of the vital sign or triage index (e.g. SpO2, ANISICCLSF_C) but preserved the first measurement (in_SpO2, in_ANISICCLSF_C). Under above condition, we did not find significant change of our model performance (logistic regression AUC 0.73, f1 0.21; voting classifier AUC 0.73, f1 0.25). Thus, we believe that the correlated features in our study did not affect the general conclusion we have made here. Thank you for your valuable suggestion.

The described model mainly uses only patient-related data and no hospital system and just one provider related factors. Using this approach can be beneficial since data collection is based on individual patients’ data only. However, patients LOS can be confounded by hospital-related factors like overcrowding, thus therefore is not a simple patient-related data point.
ANS: Thank you for your comment. We understood that LOS was affected by many factors. We have added the notice in the paragraph. Please see Line 292-293.

While I agree with the authors that prediction of URV can improve patients’ safety and decrease URVs, it’s not clear to me how prediction of URV can “avoid delays in diagnosis”.
ANS: Thank you for your comment. We have deleted the sentence to avoid unproven declare. Avoiding delays in diagnosis may be achieved if the URV prediction model was applied as an alarm system. If a physician makes a discharge decision on the patient and the alarm system showed high risk of URV, the physician may tend to perform more thorough survey to confirm the diagnosis in the first visit. Please see Line 288-290.

The authors state in the discussion that “It may also help physicians identify higher-risk patients with abdominal pain and arrange examinations earlier to confirm diagnoses or indicate appropriate outpatient department follow-up.” However, prediction of URV do not necessarily help to distinguish of URV occurred due to misdiagnosis or lack of ED discharge planning. To address this issue, the models would have to include a more differentiated assessment of URV including severity and causation of URV. While this is an interesting topic this seems to be out of scope for this article, however, the discussion should clearly distinguish that the described model can be the base for this aim, however, cannot provide this at the current state.
ANS: Thank you for your comment. We have deleted this part to avoid possibility misunderstanding. Please See Line 288-290.

Minor:

The term “free typing” is first explained in the discussion, but used within the results (e.g. Figure 3). A definition should be placed in the methods sections
ANS: Thank you for your comment. We have added the definition in Method 2.1. Please See Line 78-81.

Figures 1, 2 are blurred
ANS: Thank you for your comment. We will provide high resolution figures.

Table 3 The term “AUROC” is used in the title, however in the Abbreviation only the term “AUC” is used
ANS: Thank you for your comment. We have revised the table. Please see Table 3.

Reviewer 2 Report

Well-written paper addressing an important clinical question. Some comments regarding ML:

  1. In order to avoid data leakage, is the undersampling applied to the training set and excluded the test set?
  2. Lack of calibration: add a calibration section (calibration curves, Brier score, Hosmer-Lemeshow test)
  3. Since the prevalence of cases is low, please report the area under the Precision-Recall Curve.
  4. Did the study use textual data at all? "Free typing" should be treated as text, not a binary variable.
  5. Please report how many variables have significant missingness (>10%).
  6. Imputation should be done separately for training and test sets to avoid data leakage.
  7. Consider report Shapley value which is more informative than variable importance.
  8. Did the author include abdominal CT at the index ED visit in the models? See Healthcare (Basel)
    . 2021 Oct 30;9(11):1470. doi: 10.3390/healthcare9111470.
    Risk Factors for Early Return Visits to the Emergency Department in Patients Presenting with Nonspecific Abdominal Pain and the Use of Computed Tomography Scan
  9. Consider citing PLoS One
    . 2014 Nov 13;9(11):e112944. doi: 10.1371/journal.pone.0112944. eCollection 2014.
    Risk prediction of emergency department revisit 30 days post discharge: a prospective study
  10. ED revisits are not a valid quality measure any more. Please revise. See 
    West J Emerg Med
    . 2021 Aug 30;22(5):1124-1130. doi: 10.5811/westjem.2021.6.52212.
    Inpatient Outcomes Following a Return Visit to the Emergency Department: A Nationwide Cohort Study

Author Response

Well-written paper addressing an important clinical question. Some comments regarding ML:

  1. In order to avoid data leakage, is the undersampling applied to the training set and excluded the test set?
    ANS: Thank you for your comment. We have held back the whole test set during the train test split to prevent data leakage. Related description was added in the revised manuscript. Please See Line 103-104.

  1. Lack of calibration: add a calibration section (calibration curves, Brier score, Hosmer-Lemeshow test)
    ANS: Thank you for your comment. We have performed calibration. The Brier score and related description was added. Please See Line 126-127 and supplementary file 2.

  1. Since the prevalence of cases is low, please report the area under the Precision-Recall Curve.
    ANS: Thank you for your comment. Area Under PRC was added on the figure 5.

  1. Did the study use textual data at all? "Free typing" should be treated as text, not a binary variable.
    ANS: Thank you for your comment. It’s a binary variable. We didn’t study the textual content in this study because of some reasons, like frequent typo, since Taiwan is not English-speaking country. This feature only represents if there’s a “free typing” order or not. We hope you could understand.

  1. Please report how many variables have significant missingness (>10%).
    ANS: Thank you for your suggestion. There’re only 4 variables have significant missingness (>10%). Related description was added in the revised manuscript. Please See Line 98-101.

  1. Imputation should be done separately for training and test sets to avoid data leakage.
    ANS: Thank you for your comment. We have done it separately and related description was edited. Please see Line 107.

  1. Consider report Shapley value which is more informative than variable importance.
    ANS: Thank you for your comment. SHAP value of the Voting Classifier was calculated, and the result was showed in supplementary file 3. The result is highly similar to which we have selected before.

  1. Did the author include abdominal CT at the index ED visit in the models? See Healthcare (Basel). 2021 Oct 30;9(11):1470. doi: 10.3390/healthcare9111470. Risk Factors for Early Return Visits to the Emergency Department in Patients Presenting with Nonspecific Abdominal Pain and the Use of Computed Tomography Scan
    ANS: Thank you for your comment. We did include a variable recording if the CT was ordered or not at the index ED. Though the location of CT was not recorded, abdominal CT was most likely, considering we have included the patient with the chief complaint of abdominal pain.

  1. Consider citing PLoS One. 2014 Nov 13;9(11):e112944. doi: 10.1371/journal.pone.0112944. eCollection 2014.Risk prediction of emergency department revisit 30 days post discharge: a prospective study
    ANS: Thank you for your comment. We have cited this article. Please See Line 294.

  1. ED revisits are not a valid quality measure any more. Please revise. See West J Emerg Med. 2021 Aug 30;22(5):1124-1130. doi: 10.5811/westjem.2021.6.52212.Inpatient Outcomes Following a Return Visit to the Emergency Department: A Nationwide Cohort Study
    ANS: Thank you for your comment. We have reviewed this article and there are some differences between the article and our study. First, the severity of these 2 group patients may have significant difference. Based on clinical experience, the patients who need admission at the first visit of ED usually have severer condition comparing to the patients who could be discharged at the first visit. Second, the relationship between the total medical cost and initial ED care quality needs to be clarified. It could be confounded by the severity of patient’s disease, patient’s willing of staying in the hospital, and the patient’s social economic status. We agree that URV is not an single absolute measure for quality of care and needed to be considered with other index, but we think this article may not adequately conclude that ED URV is not a valid quality measure anymore. We still thank you for your valuable comment.

Reviewer 3 Report

This is an interesting paper describing a learning-based method to predict URVs with abdominal pain. However, I do have several methodological concerns.

  1. The data were from two sites: Taipei and Linkou. For the features used in the learning process, are there any of them with site difference? For example, blood test results may show difference between two sites due to the difference of the instrument.
  2. Feature normalization. Feature values are in different scale. If there were not normalized, the weight/coefficient of the model cannot be simply interpreted as importance.
  3. Splitting data into 80% and 20% is good. However, I recommend to do it randomly and repeat prediction for 10 times to avoid bias. 
  4.  The data is highly imbalanced. A stratified 10-fold cross-validation and training-testing parcellation would help to overcome the problem.
  5. The tuning of hyperparameters is unclear. For LR, RF and XGB, which parameters are included? 
  6. The voting weights of threes classifiers are arbitrary. Currently, the weights of each classifier is 20% to 33%. Please try a wider range of the weights, for example 5% to 90%. 
  7. Since this is clinical application, sensitivity is very important. However, current model cannot perform well to identify targets with URVs. You can modify the model to reach optimal sensitivity, instead of accuracy or AUC, during the training process.

Author Response

This is an interesting paper describing a learning-based method to predict URVs with abdominal pain. However, I do have several methodological concerns.

  1. The data were from two sites: Taipei and Linkou. For the features used in the learning process, are there any of them with site difference? For example, blood test results may show difference between two sites due to the difference of the instrument.
    ANS: Thank you for your comment. There’re some site differences, and we have revealed this part in the supplementary file. please see the supplementary file 4.

  1. Feature normalization. Feature values are in different scale. If there were not normalized, the weight/coefficient of the model cannot be simply interpreted as importance.
    ANS: Thank you for your comment. We have done the feature standardization. Please see Line 107-109.

  1. Splitting data into 80% and 20% is good. However, I recommend to do it randomly and repeat prediction for 10 times to avoid bias.
    ANS: Thank you for your comment. A stratified 10-fold cross-validation was done. Please see Line 121-123.

  1. The data is highly imbalanced. A stratified 10-fold cross-validation and training-testing parcellation would help to overcome the problem.
    ANS: Thank you for your comment. A stratified 10-fold cross-validation and training-testing parcellation were done in our method. Please see Line 121-123.

  1. The tuning of hyperparameters is unclear. For LR, RF and XGB, which parameters are included?
    ANS: Thank you for your comment. The file was added in the supplementary file. Please see supplementary file 5.

  1. The voting weights of threes classifiers are arbitrary. Currently, the weights of each classifier is 20% to 33%. Please try a wider range of the weights, for example 5% to 90%.
    ANS: Thank you for your comment. Actually, we have tried widening range and the result was the same as current one. Thank you for your suggestion

  1. Since this is clinical application, sensitivity is very important. However, current model cannot perform well to identify targets with URVs. You can modify the model to reach optimal sensitivity, instead of accuracy or AUC, during the training process.
    ANS: Thank you for your comment. Though sensitivity is important, however, focus on sensitivity may result in more numbers of false positive. Therefore, we tried to approach using a relative balance metrics such as AUC or F1 to measure the performance of models. Modifying model to reach optimal sensitivity may be very useful in certain scenario in the future. We will adjust it accordingly. Thank you for your suggestion.

Round 2

Reviewer 1 Report

Thank you for addressing some of my concerns, I am still worried about the missing data and would suggest rewriting the introduction.

The proposed alternative methods for imputation are all simple imputation methods similar to their original imputation methods, thus is it not surprisingly that results are not changed.

The authors have not addressed the type of missignissness. Considering that it is only blood pressure that is missing, it is verly likely that these data is MNAR data and should not be imputed with simple imputation methods. 

The provided supplement data are all missing figure legends.

Author Response

Reviewer 1

Thank you for addressing some of my concerns, I am still worried about the missing data and would suggest rewriting the introduction.

The proposed alternative methods for imputation are all simple imputation methods similar to their original imputation methods, thus is it not surprisingly that results are not changed.

The authors have not addressed the type of missingness. Considering that it is only blood pressure that is missing, it is very likely that these data is MNAR data and should not be imputed with simple imputation methods.

ANS: Thank you for your comment. We have been examining variables reported to have missing rate greater than 10%(They were all continuous variables: SBP1SBP2DBP1DBP2) and found that the patients with relatively mild disease tend to have those 4 missing variables. For example, those patients with relatively mild disease (triage level 3, 4; Consciousness clear; No pain killer used) would be evaluated and treated then discharged quickly to avoid ED stasis. As a result, they would not stay at ED long enough to get another blood pressure measurement. Thus, we agreed with reviewer that MNAR could possibly exist in this case. To further control this factor, we have performed senMice imputation and re-run the model (with predictive mean matching from Mice and supra parameter was set based on difference between expected values between URV and non-URV). It showed comparable results to the original one (AUC:0.73, F1: 0.26). Thus, we believe that the imputation method in our study did not affect the general conclusion we have made here. Thank you for your suggestion.

Reference: Resseguier, Noémie; Giorgi, Roch; Paoletti, Xavier Sensitivity Analysis When Data Are Missing Not-at-random, Epidemiology: March 2011 - Volume 22 - Issue 2 - p 282 doi: 10.1097/EDE.0b013e318209dec7

The provided supplement data are all missing figure legends.

ANS: Thank you for your comment. We have completed them and showed as below.

Supplementary legends

Supplementary Figure 1 Calibration curves

Supplementary Figure 2 Continuous features correlation figure matrix

Supplementary Figure 3 SHAP value of XGB

Supplementary Table 1 Features site difference

Supplementary File 1 Hyperparameters

Supplementary File 2 Weights combinations of classifiers

Reviewer 2 Report

  1. I don't think the authors answered my question regarding undersampling (one-sided selection). In order to avoid data leakage, is the undersampling applied to the training set and excluded the test set?
  2. There is no Brier score reported, nor is Hosmer-Lemeshow test. Also, please comment on the calibration curves in the text.
  3. Imputation should be done separately for training and test sets
    to avoid data leakage. Compared with the original submission, the authors seemed to just change the sentence in the revised manuscript without any re-analysis or sensitivity analysis?
  4. Please comment on the Shapley values (magnitude and direction) in the text.

Author Response

Reviewer 2

  1. I don't think the authors answered my question regarding under sampling (one-sided selection). In order to avoid data leakage, is the under sampling applied to the training set and excluded the test set?
    ANS: Thank you for your comment. In fact, we only did under sampling on training set. Under sampling was not applied to testing set. To our knowledge, under sampling is a technique to tackle imbalanced data. We did not perform under sampling on testing set. Thank you for your comment.

  1. There is no Brier score reported, nor is Hosmer-Lemeshow test. Also, please comment on the calibration curves in the text.
    ANS: Thank you for your comment. Here we reported Brier score and the results for Hosmer-Lemeshow test as suggested. Please see Line 196-201.

  1. Imputation should be done separately for training and test sets to avoid data leakage. Compared with the original submission, the authors seemed to just change the sentence in the revised manuscript without any re-analysis or sensitivity analysis?
    ANS: Thank you for your comment. We found it was just a typo in the original submission. We did imputation separately initially, so we just corrected the sentences. Thank you for your kindly remind.

  1. Please comment on the Shapley values (magnitude and direction) in the text
    ANS: Thank you for your comment. Shapley value was calculated on XGB model (supplementary figure 3), and the result revealed similar features importance as original method did, which validated the original approach is convincing. The directionality of features influence which Shapley value provided was compatible with clinical experience, for example the length of stay has negative correlation with URV and age has positive correlation with URV. Shapley value also revealed that ‘free typing’ has important contribution for prediction in some cases, which has been discussed in the original submission. Therefore, comment on the Shapley value seems repetitive in our article, but we added the results as supplementary file according to your suggestion. Thank you for your valuable suggestion.

Supplementary legends

Supplementary Figure 1 Calibration curves

Supplementary Figure 2 Continuous features correlation figure matrix

Supplementary Figure 3 SHAP value of XGB

Supplementary Table 1 Features site difference

Supplementary File 1 Hyperparameters

Supplementary File 2 Weights combinations of classifiers

Reviewer 3 Report

I suggested to mention the hyper parameters clearly in the main text, instead of in the supporting material.

And for the weights of each classifier, if the authors claimed the results are "same", please provide the results in the supplementary material.

Author Response

Reviewer 3

I suggested to mention the hyper parameters clearly in the main text, instead of in the supporting material.
ANS: Thank you for your comment. Related description was added in the revised manuscript. Please See Line 131-136.

And for the weights of each classifier, if the authors claimed the results are "same", please provide the results in the supplementary material.
ANS: Thank you for your comment. We have tried many combinations and the best combination is described in the manuscript. Please see Line 139-142 and supplementary file 2.

Supplementary legends

Supplementary Figure 1 Calibration curves

Supplementary Figure 2 Continuous features correlation figure matrix

Supplementary Figure 3 SHAP value of XGB

Supplementary Table 1 Features site difference

Supplementary File 1 Hyperparameters

Supplementary File 2 Weights combinations of classifiers

Round 3

Reviewer 1 Report

Thank you for your additional analysis. I would suggest to include the MNAR statement within the body of the manuscript.

Author Response

Thank you for your additional analysis. I would suggest to include the MNAR statement within the body of the manuscript.

ANS: Thank you for your suggestion. We have added this part into our manuscript. Please see Line 99-106.